

# The application of fractional Mel cepstral coefficient in deceptive speech detection

Xinyu Pan[1,2], Heming Zhao[1] and Yan Zhou[1]

[1] School of Electronics and Information Engineering, Soochow University, Suzhou, Jiangsu, China
[2] School of Electronics and Information Engineering, Suzhou University of Science and Technology, Suzhou, Jiangsu, China

## ABSTRACT

The inconvenience operation of EEG P300 or functional magnetic resonance imaging (FMRI) will be overcome, when the deceptive information can be effectively detected from speech signal analysis. In this paper, the fractional Mel cepstral coefficient (FrCC) is proposed as the speech character for deception detection. The different fractional order can reveal various personalities of the speakers. The linear discriminant analysis (LDA) model (which has the ability of global optimal vector mapping) is introduced, and the performance of FrCC and MFCC in deceptive detection is compared when all the data are mapped to low dimensional. Then, the hidden Markov model (HMM) is introduced as a long-term signal analysis tool. Twenty-five male and 25 female participants are involved in the experiment. The results show that the clustering effect of optimal fractional order FrCC is better than that of MFCC. The average accuracy for male and female speaker is 59.9% and 56.2%, respectively, by using the FrCC under the LDA model. When MFCC is used, the accuracy is reduced by 3.2% and 5.9%, respectively, for male and female. The accuracy can be increased to 71.0% and 70.2% for male and female speakers when HMM is used. Moreover, some individual accuracy is increased over 20%, or even more than 85%, when FrCC is introduced. The results show that the deceptive information is indeed hidden in the speech signals. Therefore, speech-based psychophysiology calculating may be a valuable research field.

# INTRODUCTION

Deception detection is regarded as an ancient and mysterious topic in the long history of human science, and there have also accumulated many useful results. The mechanism of modern polygraph is based on the changes of EEG signals, due to the contribution of psychophysiology research. Event-related potential (ERP) of P300 and functional magnetic resonance imaging for brain (fMRI) are widely used in polygraph technology (*Gao et al., 2014*; *Dong, Chen & He, 2013*), since they can intuitive reflect the process of psychological and physiological changes (such as the memories) when someone is lying. The speakers' facial expressions and postures can also be an auxiliary way to enable lie detection in some

Corresponding author
Xinyu Pan, panxy@mail.usts.edu.cn

special circumstances (*Anton et al., 2012*). The application of these techniques achieved some good results. The complex measurement process and the need for the participant's cooperation make their further promotion limited. Some information can be derived from speech, including the speaker's gender, age, emotional and mental states. Therefore, speech signals, as the carrier of deception information, may provide the basis for lie detection. Due to the complicated process from psychological activity to physiological reaction, voice-based polygraphs still exhibit the following problems. First, there is a lack of physiological experimental results demonstrating the theory basis. Second, the study of the hearing mechanism also cannot provide clear conclusions. Third, the deception detection results are not as intuitive as that of speech recognition or speaker identification, because lying is a process. Psychological stress evaluators (PSE) (*Eriksson & Lacerda, 2007*) and voice stress analyzers (VSA) (*Harnsberger et al., 2009*) are used to measure the voice tremors, which are considered as the reflection of stress. Layered voice analysis (LVA) (*Anolli & Ciceri, 1997*) claims that the link between the certain types of brain activity and the lie have been discovered. Although they are some controversial, these methods are effective to some extent. The recognition accuracy in deceptive detection is significantly low. *Bond & DePaulo (2006)* claimed that people only achieve an average of 54% correct lie-truth judgments. *Hirschberg et al. (2005)* reports a classification accuracy of 66.4% versus a chance baseline of 60.2%. A study by *Graciarena et al. (2006)* reports an accuracy of 64.0% versus a chance baseline of 60.4%. Most of the studies focused on the traditional speech features screening, and few mechanism analysis of the deceptive speech features have been made. *Christin & David (2013)* report that short-time energy, pitch trace and formant F1, F2, F3 did not show clear correlation with deceptive information. Owing to the lack of robust feature analysis, the authors still give positive prospects for speech deceptive detection.

The MFCC parameters are used as the main characteristics in the state-of-the-art speech recognition systems. The standard extraction process of MFCC makes it suitable for standard pronunciation pattern classification. Since lying varies depending on the behavior of individual differences, the standard characteristics cannot fully reflect the personality of each speaker. The conventional Fourier analysis cannot fully reveal the deceptive information hidden in the voice message. Most of the researches show the performance of GMM and SVM systems in lie detection, but there are few papers report the changes of phonetic features (including MFCC) after linear transform, and there are also very few linear classification systems used in such experiments. It is very important to evaluate the performance of some linear algorithms in feature extraction or classification, when lie detection research is in the preliminary stage. The results can provide the mathematical foundation for the further use of complex models.

In this paper, fractional Fourier transform (FrFT) is introduced in deceptive speech feature extraction, Linear Discriminant Analysis model (LDA) and hidden Markov model (HMM) are proposed for classification. Fractional Fourier transform is regarded as the angle rotation from traditional Fourier analysis. Many examples of current literature reports that the fractional Fourier transform is used for the analysis of linear frequency modulated waveform (*Pang, Liu & Shan, 2014*; *Zhu, Zhao & Tang, 2013*; *Zhu et al., 2014*).

Linear frequency modulated signals can be transformed into an impulse signal under certain angles by FrFT. The application of FrFT in the speech signal processing field is gradually increasing, and the accuracy of speech recognition and speaker identification is improved when FrFT is introduced in feature extraction (*Yin, Xie & Kuang, 2012*; *Pawan & Raghunath, 2013*). The fractional order linear canonical transform algorithm also obtains a good result in speech signal reconstruction (*Qiu, Li & Li, 2013*). There are many successful applications with LDA or HMM in the field of speech signal processing. Behzad used LDA and its modified algorithm to reduce the speech recognition error rate (*Behzad et al., 2011*), and Ana and Jordi used LDA to achieve the phonetic features analysis of snoring patients (*Ana et al., 2014*; *Jordi et al., in press*). *Rabiner & Schafer (2007)* and *Matthias & John (2009)* successfully used HMM in speech recognition.

In this paper, the Mel cepstral coefficient in fractional domain (called Fractional Mel Cepstral Coefficient, FrCC) is extracted for voice spectrum analysis. Then, LDA and HMM models are used to distinguish between truth and deception. Different fractional order spectrum analysis in Mel domain can further refine the pronunciation characteristics of all liars. The rotation of MFCC parameters in the Mel domain is introduced, and the details of each individual difference in speech signal may reinforce to a certain extent. The LDA algorithm can help us to find the best rotation angle and obtain the best division result. The HMM-based time series model can reveal the psychological and physiological changes when lying, and increase the recognition accuracy. The FrCC parameter corresponding with the highest accuracy can be called optimal order FrCC. The deceptive detection accuracy of all optimal order FrCC is higher than that of MFCC, so the acoustic characteristics of speech signals can provide some support for lie detection.

The arrangement of the full text is as follows. The first section introduces the study of distinguish features of deceptive speech, and provides the application basis of fractional Fourier analysis in this subject. The second section presents the calculation of Fractional Mel cepstral coefficients. The LDA algorithm and HMM model are introduced in the third section. Experiment results and analysis are described in the fourth section. The conclusions are drawn finally.

# DECEPTIVE SPEECH FEATURE INTRODUCTION AND FRACTIONAL FOURIER TRANSFORM APPLICATION FOUNDATION

## The introduction of speech signals based polygraph application

Lying is a process which moves from psychological activity to physiological execution. Firstly, people decide to lie from their conscious mind, and then organize the language to cover the real content. Finally, control vocal organs to form the voice. *Tommaso, Fabio & Massimo (2013)* studied the deception detection results among the people of different personality by the means of pattern recognition. The conclusion is that the outgoing personality groups are easier to be identified. It is also proved that lie detection has individual differences. *Lamb & Skillicorn (2013)* reported that the frequency of words with different parts of speech appear in the trial process may also be a way to analyze the possibility

of lying. In the field of natural language processing, *Christie, David & Dursun (2008)* showed that the result is influenced by lexical, syntax, sentence length and motivation. Furthermore, the organization of text and voice can be used in an anti-phishing detection system, preventing people from being cheated in instant messengers (*Mohammed & Lakshmi, 2012*). These researchers discuss the deceptive speech detection results in psychology and natural language processing fields, including personality difference, pronunciation difference, language expression, sentence organization and so on. Then, *Sofia et al. (2013)* researched the speaker's differences under normal state and tense state. *Cheryl et al. (2013)* summarized that the person under tension will show the following case: increase of adrenaline, higher blood pressure, sympathetic excitement, bronchiectasis, and cricothyroid muscle tension. Some people tended to show increased pitch frequency and voice trembling, but not everyone exhibited these traits. These physiological studies have proved the existence of the difference between normal and deceptive state, and provide some simple basis in the physiological field for speech deception detection.

The above conclusion from psychology, linguistics and physiological aspects are relatively consistent, but the research in the acoustic field have different results. *Gopalan & Wenndt (2007)* claim that the trace of pitch and first formant, which are processed by AM-FM model and Teager operator, presented the definite difference between normal and deceptive speech. However, *Christin & David (2013)* gave the opposite conclusion. They executed several experiments, and the results are presented on a range of speech parameters including fundamental frequency, overall amplitude and mean vowel formants F1, F2, F3. They could not establish a significant correlation between deceptiveness and truthfulness; the two results appeared opposite. Pitch and formants are susceptible to the influence of speaker's pronunciation habit, language content, and coarticulation. The parameters will also be changed due to different extraction algorithm, so it is not a good choice for using them as the speech features for lie detection. There is little research to reveal the process from psychological activity to speech production. *Muhammad & Kaliappan (2013)* use the Bark spectrum as speech features, and use the neural network as the classification model to identify the truthfulness and deceptiveness. Therefore, using robust acoustic characteristics for deceptive speech identification should be more reliable, and there is still plenty of scope for more progress.

## The fractional Fourier transform application foundations

Current research has not investigated the feature difference between normal speech and the lie. Physiological studies also could not provide any explanation for whether there are specific changes of articulators or not when people are lying. The existing information is only obtained from speech features statistical research result or classification conclusion by traditional pattern recognition models. The speech features in the frequency domain are mostly achieved by power spectrum transformation. If the liar's psychological and physiological changes indeed impact the frequency of speech, the deceptive information can be extracted by short-time frequency analysis for voice signals. But it does not work, if only the speech phase is changed. Therefore, a speech feature which can express both the

change of frequency and phase is needed. The fractional Fourier analysis is applicable to the task.

That being the case we take the cosine signal as an example to compare the difference between the traditional Fourier transform and fractional Fourier transform. (Please refer to the next section for the detail description of FrFT.)

$$x(t) = \cos(\varpi_0 t) \quad \Leftrightarrow \quad |X(\varpi)| = \pi\left(\delta(\varpi - \varpi_0) + \delta(\varpi + \varpi_0)\right) \tag{1}$$

$$x(t) = \cos(\varpi_0 t + \theta) \quad \Leftrightarrow \quad |X(\varpi)| = \pi\left(\delta(\varpi - \varpi_0) + \delta(\varpi + \varpi_0)\right). \tag{2}$$

Through (1) and (2), it is shown that the Fourier amplitude–frequency response can't reflect the phase difference of the two signals. According to trigonometric formula, Eq. (2) can be expanded and transformed by FrFT as the Eq. (3).

$$x(t) = \cos(\varpi_0 t + \theta) = \cos(\varpi_0 t)\cos(\theta) - \sin(\varpi_0 t)\sin(\theta)\ldots$$

$$\overset{frft}{\Leftrightarrow} \quad X_\alpha(u) = \cos(\theta)X_\alpha[\cos(\varpi_0 t)](u) - \sin(\theta)X_\alpha[\sin(\varpi_0 t)](u). \tag{3}$$

Then:

$$X_\alpha[\cos(\varpi_0 t)](u) = \sqrt{1 + j\tan(\alpha)}\cos(u\varpi_0\sec(\alpha))\exp\left(-\frac{j}{2}(\varpi_0^2 + u^2)\tan(\alpha)\right) \tag{4}$$

$$X_\alpha[\sin(\varpi_0 t)](u) = \sqrt{1 + j\tan(\alpha)}\sin(u\varpi_0\sec(\alpha))\exp\left(-\frac{j}{2}(\varpi_0^2 + u^2)\tan(\alpha)\right) \tag{5}$$

$$\begin{aligned}
X_\alpha(u) &= \cos(\theta)\sqrt{1 + j\tan(\alpha)}\cos(u\varpi_0\sec(\alpha))\exp\left(-\frac{j}{2}(\varpi_0^2 + u^2)\tan(\alpha)\right)\ldots \\
&\quad - \sin(\theta)\sqrt{1 + j\tan(\alpha)}\sin(u\varpi_0\sec(\alpha))\exp\left(-\frac{j}{2}(\varpi_0^2 + u^2)\tan(\alpha)\right) \\
&= \sqrt{1 + j\tan(\alpha)}\exp\left(-\frac{j}{2}(\varpi_0^2 + u^2)\tan(\alpha)\right)(\cos(\theta)\cos(u\varpi_0\sec(\alpha)) \\
&\quad + \sin(\theta)\sin(u\varpi_0\sec(\alpha))) \\
&= \sqrt{1 + j\tan(\alpha)}\exp\left(-\frac{j}{2}(\varpi_0^2 + u^2)\tan(\alpha)\right)\cos(u\varpi_0\sec(\alpha) + \theta). 
\end{aligned} \tag{6}$$

Equation (6) expressed the FrFT result of $\cos(\varpi_0 t + \theta)$. The phase $\theta$ still exists in $|X_\alpha(u)|$, so Eq. (6) can reserve the phase information.

The use of speech signal for lie detection is only in the preliminary stages. If the difference between truthfulness and deceptiveness is really expressed by the amplitude and phase of speech spectrum, the fractional Fourier transform analysis method should be an effective way to reveal the distinction. So as to suit for the diversity of speakers, variety orders of FrFT should be involved. The difference between normal speech and lie can be enhanced due to some orders of FrFT.

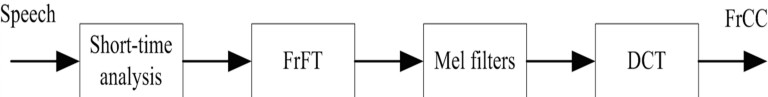

**Figure 1 FrCC extraction process.**

# FRACTIONAL MEL CEPSTRAL COEFFICIENT (FrCC) EXTRACTION

The FrCC parameters are modified based on MFCC. First step is short-time analysis, then transform time domain samples to frequency domain by FrFT under a set of rotation angles. The following step is to divide the signals into Mel frequency domain by triangular filters, then conduct by a DCT at last. So the whole process is shown in Fig. 1.

The details of FrCC calculation steps are shown as follows:

(A) The fractional Fourier transform for speech signals is shown in (7).

$$S_\alpha(u) = F_p[s(t)] = \int_{-\infty}^{+\infty} s(t)K_\alpha(t,u)dt \tag{7}$$

Here, $\alpha = p\frac{\pi}{2}$, $p$ is the set of real numbers, the order of FrFT. $K_\alpha(t,u)$ is the primary function of FrFT, and its specific expressions is presented in Eq. (8).

$$K_\alpha(t,u) = \begin{cases} \sqrt{\dfrac{1-j\cot\alpha}{2\pi}}\exp\left(j\dfrac{t^2+u^2}{2}\cot\alpha - jtu\csc\alpha\right), & \alpha \neq n\pi \\ \delta(t-u), & \alpha = 2n\pi \\ \delta(t+u), & \alpha = (2n\pm1)\pi. \end{cases} \tag{8}$$

According to the properties of (8), when $\alpha = \frac{\pi}{2}$ the fractional Fourier transform is equal to the traditional Fourier transform.

(B) Equation (9) provides the spectrum mapping operator from fractional domain to Mel frequency domain.

$$M(u) = 1125\ln(1+u/700). \tag{9}$$

The Mel frequency band is based on the human auditory characteristics, it should also meet such requirement in fractional domain. Therefore, Eq. (10) presents the frequency projection operator.

$$u = f \times \sin\alpha \quad (f \text{ is linear frequency}). \tag{10}$$

The output of each triangular filter is $|S_\alpha^M(u)|$, $M$ refers to $M$-th Mel component. Figure 2 shows the fractional Mel frequency schematic.

(C) The fractional Mel cepstral coefficients (FrCC) can be achieved after a DCT of $|S_\alpha^M(u)|$.

$$\text{FrCC}_n = \sqrt{\frac{2}{N}}\sum_{k=1}^{M}\text{Log}|S_\alpha^M(u)|\cos[\pi(k-0.5)n/M]. \tag{11}$$

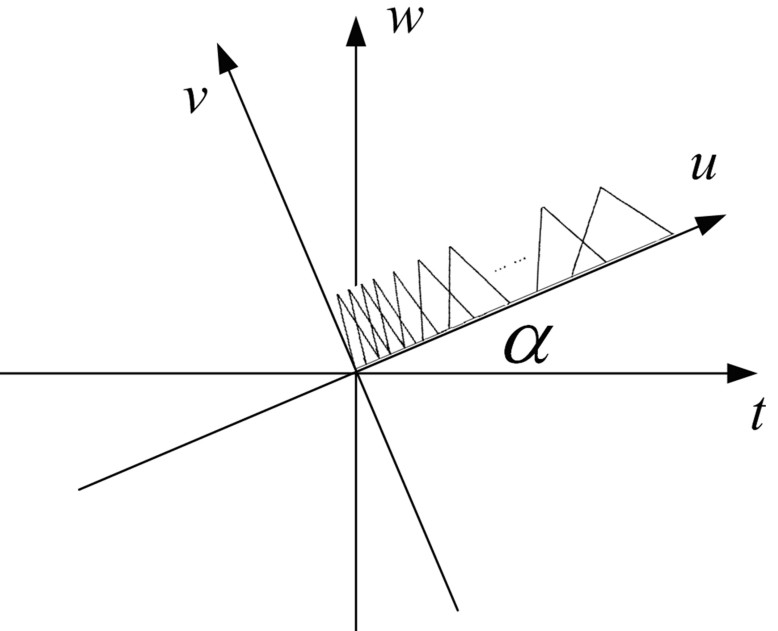

**Figure 2** The fractional Mel triangular filter schematic.

The FrCC parameters can be calculated according to the above equations. And FrCC is equal to MFCC when $\alpha = \frac{\pi}{2}$.

# LDA AND HMM MODEL

## LDA algorithm

The main function of linear discriminant analysis (LDA) is to project the high-dimensional samples onto a low dimensional space. It is aimed to maximize the distance between classes, and minimize the distance in the class. Therefore, LDA is suitable for two group classification task, such as truthfulness and deceptiveness division. The $S = \{s_1, s_2, \ldots, s_n\}$ refers to the training voice set, and $s_i$ is belong to $\omega_1$ or $\omega_2$, which represents for normal speech or lie respectively. A projection operator $w$ defined as the best vector may map $x_i$ to the one-dimensional $y$.

$$y = w^T s_i. \tag{12}$$

Then, it is very easy to make a decision by a simple comparison.

$$\begin{cases} s_i \in \omega_1 & y \geq t \\ s_i \in \omega_2 & y < t \end{cases} \quad (t \text{ is a threshold}). \tag{13}$$

So the main task of LDA is to calculate the optimal mapping vector $w$.

The mean of each category can be expressed as

$$\mu_i = \frac{1}{N_i} \sum_{s \in \omega_i} s_i. \tag{14}$$

The mean value is changed after projection.

$$\tilde{\mu}_i = \frac{1}{N_i} \sum_{y \in \omega_i} y = \frac{1}{N_i} \sum_{s \in \omega_i} w^T s = w^T \mu_i. \tag{15}$$

The distance between two means is

$$D(w) = |\tilde{\mu}_1 - \tilde{\mu}_2| = |w^T(\mu_1 - \mu_2)|. \tag{16}$$

And the variance after projection is

$$\tilde{d}_i^2 = \sum_{y \in \omega_i} (y - \tilde{\mu}_i)^2. \tag{17}$$

Define the objection function $J(w)$, when reaching the max ratio of the distance between these two categories and the variance within the classes, the $w$ is the best vector.

$$J(w) = \frac{|\tilde{\mu}_1 - \tilde{\mu}_2|^2}{\tilde{d}_1^2 + \tilde{d}_2^2}. \tag{18}$$

The Eq. (17) can be decomposed into (19).

$$\tilde{d}_i^2 = \sum_{y \in \omega_i} (y - \tilde{\mu}_i)^2 = \sum_{s \in \omega_i} (w^T s - w^T \mu_i)^2 = \sum_{s \in \omega_i} w^T (x - \mu_i)(x - \mu_i)^T w. \tag{19}$$

Then define

$$d_i = \sum_{s \in \omega_i} (s - \mu_i)(s - \mu_i)^T. \tag{20}$$

And we may have

$$d_w = d_1 + d_2. \tag{21}$$

So

$$\tilde{d}_1^2 + \tilde{d}_2^2 = w^T d_w w. \tag{22}$$

And

$$(\tilde{\mu}_1 - \tilde{\mu}_2)^2 = (w^T \mu_1 - w^T \mu_2)^2 = w^T(\mu_1 - \mu_2)(\mu_1 - \mu_2)^T w = w^T d_k w. \tag{23}$$

The Eq. (18) can be written as

$$J(w) = \frac{w^T d_k w}{w^T d_w w}. \tag{24}$$

The optimal vector is

$$w = d_w^{-1}(\mu_1 - \mu_2). \tag{25}$$

The test voice set can be easily divided into two groups by Eqs. (12) and (13) when the $w$ is obtained.

## Hidden Markov model

The hidden Markov model (HMM) can be considered as a generalization of a mixture model. The hidden variables are related through a Markov process, and the observation is controlled by the hidden state. The state is not directly visible in a HMM, but output observation is visible and dependent on the state. Each state has a probability distribution corresponding to the possible output. Therefore, the output sequence generated by an HMM presents some information about the sequence of invisible states.

The random variable $x(t)$ presents the hidden state at time $t$ (with $x(t) \in \{x_1, x_2, x_3\}$). The random variable $y(t)$ is considered as the speech observation at time $t$ (with $y(t) \in \{y_1, y_2, y_3, y_4\}$). According to the basic theory of Markov process, it is clear that the conditional probability distribution of the hidden variable $x(t)$ only depends on the value of the $x(t-1)$. The values at time $t-2$ and before have no influence. The value of the speech observation $y(t)$ also depends on the value of the $x(t)$.

Two types of parameters, called transition probabilities and output probabilities, are contained in a HMM. The hidden state at time $t$ is determined by hidden state at time $t-1$ according to the transition probabilities. There is also a set of output probabilities to describe the distribution of the observed variable.

There are some important parameters in a HMM.

1. $N$, the number of states in the model.
2. $M$, the number of observation symbols per state.
3. The transition probability distribution.

$$a_{ij} = P(x_{t+1} = s_j | x_t = s_i). \tag{26}$$

4. The output observation probability distribution.

$$b_j(k) = P(y_t = o_k | x_t = s_j). \tag{27}$$

5. The initial state distribution.

$$\pi_i = P(x_t = s_i). \tag{28}$$

The famous forward-backward algorithm, EM algorithm, and Viterbi algorithm can be used to train the models and solve the recognition task (*Rabiner & Schafer, 2007*; *Matthias & John, 2009*). Although the HMM is a traditional method used in recognition systems, it is still a suitable model in deceptive speech detection with time series speech signals.

# EXPERIMENT RESULTS AND ANALYSIS

## Speech database

The liar's appearance has a direct relationship with individual personality, culture background, conversation content and the cost of being seen through the lie. Therefore,

the speech sources should be collected in a real circumstance. According to the concealed information test theory (*Verschuere, Ben-shakhar & Meijer, 2011*), we designed an interesting game, and the speech database is selected from the game records. There are two groups in the game, which are called A-group and B-group. Every person in A-group should tell a story, and the persons in B-group can ask all kinds of questions according to the story. Due to the different stories told by peoples in A-group, the questions and answers are different from each other. Since the persons in B-group do not know whether the story is experienced the storyteller himself/herself, they should decide the true or false through the teller's answers. If the persons in B-group speculate the correct result, they win the game and obtain some reward. Otherwise, the person in A-group wins. So if the story is a lie, the teller should try his/her best to keep the secret from every one to win the game. People in B-group should ask as many questions as possible (generally more than 10 questions) to make the liars nervous and make mistakes. Here, we select the every fake stories and fake answers as the deceptive speech samples. Then the corresponding tellers should record a set of normal speech in a calm environment, the topics can include such as self-introduction, hobbies, and daily life topics and so on. These records should long enough to cover as much as possible syllables in Mandarin Chinese. So the normal speech samples are collected.

At last, we reserved useful records of 50 participants, including the 25 men and 25 women. Due to some limitations, the participants are mainly 25 to 35 years old. The SNR of all the samples are more than 25dB. All speech is mono sampled at 8 kHz and quantified with 16 bits. The frame length is 20 ms, and the overlap is 10 ms when the speech is under short-time analysis. The data set is divided into two parts, namely the training set and test set. The experiment is under a unified standard to divide the data set due to the different length of the every people's speech sample. The training set contains 30% of whole record, and the remaining data are regarded as the test set.

**Human Ethics**

This research was approved by the Institutional Review Boards of Soochow University School of Electronics and Information Engineering, and Suzhou University of Science and Technology School of Electronics and Information Engineering. The speech set recording is carried out in a game style, so all the participants are confirmed with verbal consent.

## The experiment results for LDA model

The experiment step is expressed as follows:

(A) Divide the speech signals into short frames with the length of 20 ms and overlap of 10 ms.

(B) Extracted the FrCC parameters from every frame, and 12 FrCC coefficients and 12 delta FrCC coefficients are used as the FrCC vector from one speech frame. The range of angle is $\alpha \in (0, \pi)$, with the $0.01\pi$ as the step. So there are 100 FrCC vector groups in every frame.

(C) Select 30% of the total data as the train set, and use LDA algorithm to calculate the optimal vector $w$.

**Table 1  Accuracy of men set for LDA model.**

| People ID | 1 | 2 | 3 | 4 | 5 | 6 | 7 | 8 | 9 | 10 | 11 | 12 |
|---|---|---|---|---|---|---|---|---|---|---|---|---|
| $\alpha$ ($\times \pi$) | 0.83 | 0.70 | 0.08 | 0.90 | 0.52 | 0.52 | 0.88 | 0.49 | 0.62 | 0.50 | 0.37 | 0.52 |
| FrCC Accuracy (%) | 56.8 | 65.5 | 49.5 | 54.4 | 69.8 | 59.7 | 65.6 | 62.2 | 53.2 | 60.3 | 47.4 | 45.5 |
| MFCC Accuracy (%) | 48.0 | 62.1 | 49.1 | 53.6 | 67.7 | 59.2 | 61.6 | 61.9 | 50.5 | 60.3 | 46.1 | 45.4 |

| People ID | 13 | 14 | 15 | 16 | 17 | 18 | 19 | 20 | 21 | 22 | 23 | 24 | 25 |
|---|---|---|---|---|---|---|---|---|---|---|---|---|---|
| $\alpha$ ($\times \pi$) | 0.01 | 0.99 | 0.52 | 0.15 | 0.20 | 0.34 | 0.51 | 0.51 | 0.02 | 0.50 | 0.48 | 0.97 | 0.51 |
| FrCC Accuracy (%) | 58.9 | 60.9 | 73.4 | 56.0 | 68.2 | 60.5 | 65.9 | 65.2 | 57.6 | 58.6 | 56.5 | 60.1 | 65.4 |
| MFCC Accuracy (%) | 52.1 | 51.9 | 72.6 | 36.7 | 59.3 | 50.8 | 65.7 | 65.1 | 52.5 | 58.6 | 55.5 | 54.8 | 65.3 |

**Table 2  Accuracy of women set for LDA model.**

| People ID | 1 | 2 | 3 | 4 | 5 | 6 | 7 | 8 | 9 | 10 | 11 | 12 |
|---|---|---|---|---|---|---|---|---|---|---|---|---|
| $\alpha$ ($\times \pi$) | 0.65 | 0.77 | 0.96 | 0.97 | 0.75 | 0.97 | 0.48 | 0.50 | 0.60 | 0.57 | 0.77 | 0.44 |
| FrCC Accuracy (%) | 61.5 | 56.0 | 52.3 | 53.1 | 68.7 | 55.2 | 55.0 | 59.1 | 62.7 | 57.4 | 70.1 | 56.4 |
| MFCC Accuracy (%) | 59.8 | 47.3 | 39.0 | 36.6 | 52.5 | 41.6 | 53.7 | 59.1 | 61.7 | 57.2 | 58.4 | 51.9 |

| People ID | 13 | 14 | 15 | 16 | 17 | 18 | 19 | 20 | 21 | 22 | 23 | 24 | 25 |
|---|---|---|---|---|---|---|---|---|---|---|---|---|---|
| $\alpha$ ($\times \pi$) | 0.51 | 0.52 | 0.21 | 0.36 | 0.09 | 0.99 | 0.99 | 0.51 | 0.45 | 0.01 | 0.68 | 0.51 | 0.60 |
| FrCC Accuracy (%) | 39.9 | 60.1 | 52.6 | 57.7 | 64.3 | 56.6 | 54.7 | 43.3 | 57.0 | 49.9 | 47.8 | 68.6 | 46.3 |
| MFCC Accuracy (%) | 39.8 | 60.0 | 47.0 | 50.4 | 49.1 | 55.0 | 42.2 | 41.7 | 56.5 | 44.6 | 43.3 | 68.6 | 40.8 |

(D) Take the remaining 70% of data as test set. Map the test set into low-dimensional space by $w$.

Then use Eqs. (12) and (13) to make the decision and the statistical accuracy can be obtained at last.

In this experiment, the recognition results of MFCC parameters are taken as a benchmark to compare with that of FrCC parameters. The results are shown in Tables 1, 2 and Figs. 3, 4. The first line of these tables denotes the people's ID. The second line indicates the corresponding $\alpha$ of highest accuracy for FrCC. The accuracy of FrCC and MFCC are shown in line 3 and 4, respectively.

In order to further refine the improvement of FrCC, the vector variance is introduced to compare the clustering performance of the two parameters. The vector variance is shown in Eqs. (29) and (30).

$$R_1 = \frac{\sum_{i=1}^{N}(normfrcc_i - \overline{normfrcc})^2}{\sum_{i=1}^{N}(normmfcc_i - \overline{normmfcc})^2} \tag{29}$$

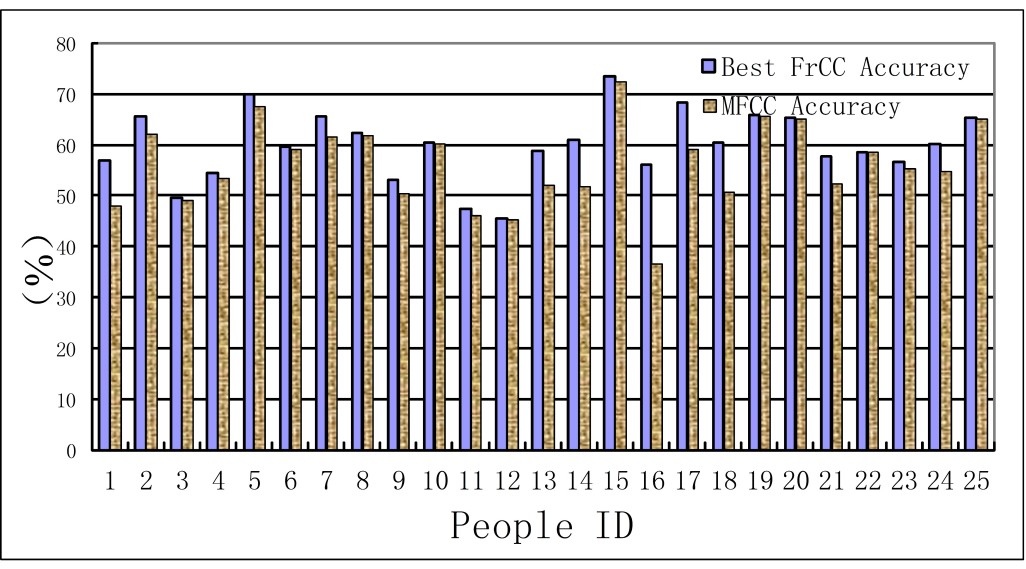

**Figure 3** Accuracy of men set under LDA model.

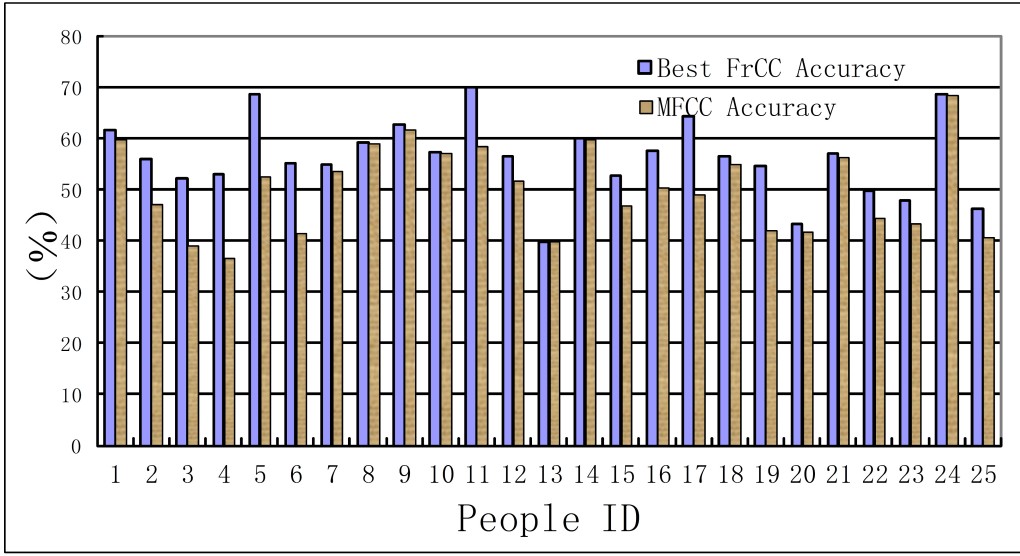

**Figure 4** Accuracy of women set under LDA model.

$$R_2 = \frac{\sum_{j=1}^{M}(decfrcc_j - \overline{decfrcc})^2}{\sum_{j=1}^{M}(decmfcc_j - \overline{decmfcc})^2}. \tag{30}$$

Here, the $normfrcc_i$ presents the FrCC of normal speech, and the $\overline{normfrcc}$ presents the mean vector. The $normmfcc_i$ and $\overline{normmfcc}$ present the MFCC of normal speech and MFCC mean vector, respectively. The $R_1$ in Eq. (29) denotes the vector variance ratio

**Table 3  Vector variance ratio of men set.**

| People ID | 1 | 2 | 3 | 4 | 5 | 6 | 7 | 8 | 9 | 10 | 11 | 12 |
|---|---|---|---|---|---|---|---|---|---|---|---|---|
| $\alpha\ (\times\pi)$ | 0.83 | 0.70 | 0.08 | 0.90 | 0.52 | 0.52 | 0.88 | 0.49 | 0.62 | 0.50 | 0.37 | 0.52 |
| $R_1$ | 0.75 | 0.79 | 0.55 | 0.55 | 0.99 | 1.01 | 0.63 | 0.99 | 0.87 | 1.00 | 0.70 | 0.98 |
| $R_2$ | 0.74 | 0.70 | 0.53 | 0.62 | 1.00 | 0.97 | 0.66 | 0.99 | 0.84 | 1.00 | 0.72 | 0.97 |

| People ID | 13 | 14 | 15 | 16 | 17 | 18 | 19 | 20 | 21 | 22 | 23 | 24 | 25 |
|---|---|---|---|---|---|---|---|---|---|---|---|---|---|
| $\alpha\ (\times\pi)$ | 0.01 | 0.99 | 0.52 | 0.15 | 0.20 | 0.34 | 0.51 | 0.51 | 0.02 | 0.50 | 0.48 | 0.97 | 0.51 |
| $R_1$ | 0.60 | 0.59 | 0.99 | 0.69 | 0.70 | 0.74 | 0.99 | 0.99 | 0.57 | 1.00 | 1.01 | 0.71 | 1.00 |
| $R_2$ | 0.63 | 0.50 | 0.98 | 0.64 | 0.67 | 0.67 | 1.00 | 0.98 | 0.52 | 1.00 | 1.01 | 0.59 | 0.99 |

**Table 4  Vector variance ratio of women set.**

| People ID | 1 | 2 | 3 | 4 | 5 | 6 | 7 | 8 | 9 | 10 | 11 | 12 |
|---|---|---|---|---|---|---|---|---|---|---|---|---|
| $\alpha\ (\times\pi)$ | 0.65 | 0.77 | 0.96 | 0.97 | 0.75 | 0.97 | 0.48 | 0.50 | 0.60 | 0.57 | 0.77 | 0.44 |
| $R_1$ | 0.82 | 0.68 | 0.59 | 0.49 | 0.66 | 0.54 | 0.99 | 1.00 | 0.98 | 0.98 | 0.61 | 0.96 |
| $R_2$ | 0.81 | 0.66 | 0.49 | 0.54 | 0.66 | 0.57 | 0.98 | 1.00 | 0.89 | 0.99 | 0.63 | 0.94 |

| People ID | 13 | 14 | 15 | 16 | 17 | 18 | 19 | 20 | 21 | 22 | 23 | 24 | 25 |
|---|---|---|---|---|---|---|---|---|---|---|---|---|---|
| $\alpha\ (\times\pi)$ | 0.51 | 0.52 | 0.21 | 0.36 | 0.09 | 0.99 | 0.99 | 0.51 | 0.45 | 0.01 | 0.68 | 0.51 | 0.60 |
| $R_1$ | 0.99 | 0.99 | 0.67 | 0.79 | 0.48 | 0.54 | 0.46 | 1.00 | 0.94 | 0.43 | 0.72 | 0.99 | 0.84 |
| $R_2$ | 0.99 | 0.99 | 0.67 | 0.76 | 0.48 | 0.56 | 0.44 | 0.98 | 0.96 | 0.45 | 0.69 | 0.99 | 0.84 |

between FrCC and MFCC of normal speech. The $R_2$ in Eq. (30) denotes the vector variance ratio between FrCC and MFCC of deceptive speech. The results are presented in Tables 3 and 4.

## The experiment results of HMM model

There are many sophisticated tools for HMM training and testing, such as HTK or Matlab software package. The speech signals should also be divided into short frames. Then the speech characters such as FrCC and MFCC can be regarded as the observations, the psychophysiology status is regarded as the invisible states. The 30% of the total data is regarded as the train set, and the remaining data is the test set. The speech characters changed frame to frame, and the hidden Markov chain can present the process of the psychophysiology changes. The experiment results are shown in Tables 5 and 6. The comparison between the men set and women set are in Figs. 5 and 6.

## Results analysis and discussion

In the sections 'The experiment results for LDA model' and 'The experiment results of HMM model,' the experiment results show that the identification accuracy of FrCC parameters under certain angles is higher than that of MFCC parameters. The FrCC coefficients introduced to the LDA model make the clustering performance much better.

**Table 5  Accuracy of men set for HMM.**

| People ID | 1 | 2 | 3 | 4 | 5 | 6 | 7 | 8 | 9 | 10 | 11 | 12 |
|---|---|---|---|---|---|---|---|---|---|---|---|---|
| $\alpha$ ($\times\pi$) | 0.83 | 0.70 | 0.08 | 0.90 | 0.52 | 0.52 | 0.88 | 0.49 | 0.62 | 0.50 | 0.37 | 0.52 |
| FrCC Accuracy (%) | 67.1 | 73.3 | 59.9 | 70.1 | 81.1 | 72.3 | 77.9 | 74.2 | 60.0 | 74.7 | 61.9 | 64.0 |
| MFCC Accuracy (%) | 52.3 | 68.2 | 57.8 | 63.4 | 77.5 | 64.0 | 70.6 | 70.0 | 58.8 | 74.7 | 63.3 | 61.7 |

| People ID | 13 | 14 | 15 | 16 | 17 | 18 | 19 | 20 | 21 | 22 | 23 | 24 | 25 |
|---|---|---|---|---|---|---|---|---|---|---|---|---|---|
| $\alpha$ ($\times\pi$) | 0.01 | 0.99 | 0.52 | 0.15 | 0.20 | 0.34 | 0.51 | 0.51 | 0.02 | 0.50 | 0.48 | 0.97 | 0.51 |
| FrCC Accuracy (%) | 75.5 | 72.8 | 82.0 | 67.7 | 79.5 | 70.0 | 73.2 | 69.6 | 63.7 | 73.0 | 61.2 | 72.8 | 77.9 |
| MFCC Accuracy (%) | 66.0 | 65.4 | 80.1 | 52.0 | 71.0 | 64.1 | 72.9 | 69.6 | 60.1 | 73.0 | 60.4 | 63.3 | 78.7 |

**Table 6  Accuracy of women set for HMM.**

| People ID | 1 | 2 | 3 | 4 | 5 | 6 | 7 | 8 | 9 | 10 | 11 | 12 |
|---|---|---|---|---|---|---|---|---|---|---|---|---|
| $\alpha$ ($\times\pi$) | 0.65 | 0.77 | 0.96 | 0.97 | 0.75 | 0.97 | 0.48 | 0.50 | 0.60 | 0.57 | 0.77 | 0.44 |
| FrCC Accuracy (%) | 74.1 | 66.3 | 65.9 | 68.2 | 81.9 | 64.5 | 67.7 | 70.2 | 77.6 | 63.4 | 85.4 | 65.7 |
| MFCC Accuracy (%) | 68.2 | 59.0 | 60.1 | 66.0 | 70.5 | 63.2 | 59.5 | 70.2 | 73.1 | 61.9 | 69.9 | 59.2 |

| People ID | 13 | 14 | 15 | 16 | 17 | 18 | 19 | 20 | 21 | 22 | 23 | 24 | 25 |
|---|---|---|---|---|---|---|---|---|---|---|---|---|---|
| $\alpha$ ($\times\pi$) | 0.51 | 0.52 | 0.21 | 0.36 | 0.09 | 0.99 | 0.99 | 0.51 | 0.45 | 0.01 | 0.68 | 0.51 | 0.60 |
| FrCC Accuracy (%) | 71.1 | 72.1 | 67.7 | 68.1 | 79.4 | 63.2 | 68.0 | 70.2 | 66.6 | 69.0 | 68.5 | 80.3 | 59.1 |
| MFCC Accuracy (%) | 70.5 | 70.0 | 66.6 | 62.1 | 68.8 | 62.8 | 58.6 | 71.7 | 65.9 | 57.4 | 53.9 | 80.3 | 55.7 |

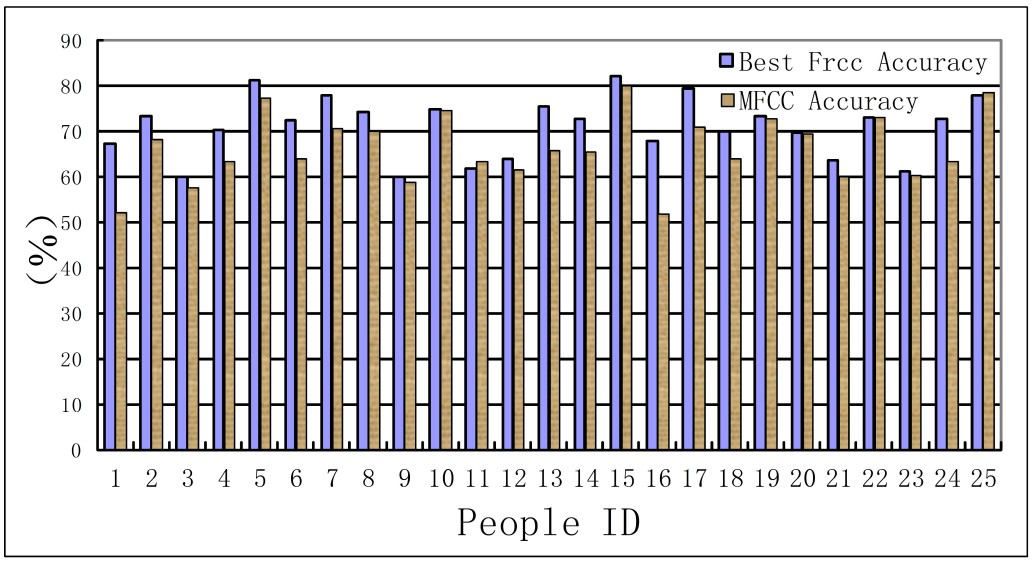

**Figure 5  Accuracy of men set for HMM model.**
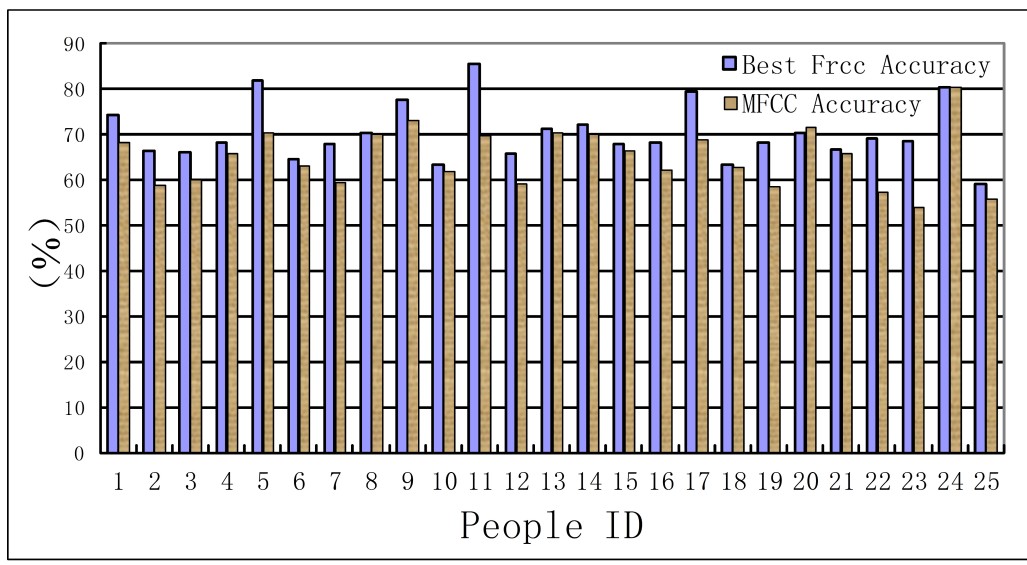

**Figure 6 Accuracy of women set for HMM model.**

The accuracy will be increased when HMM model is used to enhance the contextual information. The following paragraphs give some brief explanation to the experiment results.

(A) In the LDA recognition system, the men groups' average accuracy of FrCC with best angle is 59.9%, and MFCC is 56.3%. The average of best angle is $\overline{\alpha} = 0.51\pi$, and the variance of $\alpha$ is $D(\alpha) = 0.23\pi$. The women groups' average accuracy of FrCC with best angle is 56.2%, and MFCC is 50.3%. The average of best angle is $\overline{\alpha} = 0.59\pi$, and the variance of $\alpha$ is $D(\alpha) = 0.22\pi$. The best angle of 10, 22 in men's group and 8 in women's group is $\frac{\pi}{2}$. In these cases, the FrCC coefficients are equal to MFCC coefficients. In the other cases, the identification performance of FrCC under LDA model is better than that of MFCC. The accuracy increased from 36.7% to 56.0% when FrCC is introduced in the 16th men. And the accuracy of many people is increased over 10%. Due to individual differences, the accuracy is only a little increased with some people. Overall, when FrCC coefficients are involved, the average accuracy is increased by 3.6% in men group and 5.9% in women group, respectively. Although the average growth of accuracy is not very large, there was great progress with some individuals. The FrCC parameters can therefore improve the deceptive detection performance.

(B) The performance of FrCC parameters may be changed with different $\alpha$ of FrFT. Due to the diversity and non-stationary characteristics of speech, and personality difference of the speakers, it is impossible to determine the optimal $\alpha$ before the experiment. The best $\alpha$ is selected by the highest accuracy. So the mechanism of selection algorithm should be further studied.

(C) Most of the $R_1$ and $R_2$ is less than 1 in Tables 3 and 4. It is shown that the clustering performance of certain FrCC is much better than that of MFCC. The FrCC data is concentrated to the clustering center. The existence of $\alpha$ enhanced the appearance of
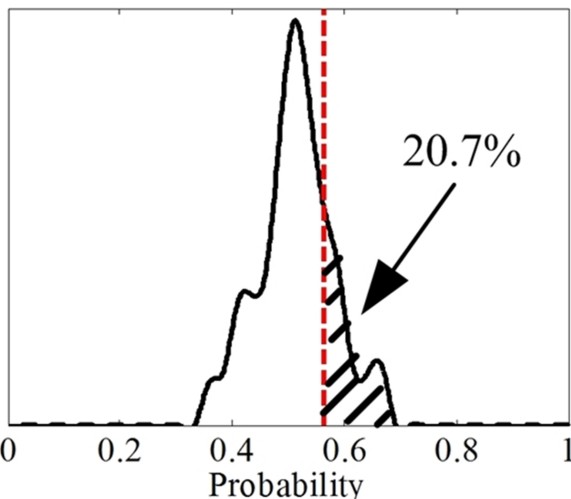

**Figure 7 The distribution of FrCC based deceptive speech recognition accuracy by LDA in men set.**

the change of articulator, when people are lying or under stress. These faint details could not be reflected by MFCC. These conclusions may be explained by phase change statement described in section 'The Fractional Fourier Transform application foundations,' and should be further verified by physiology research.

(D) The experiment results show that the performance of FrCC is better than that of MFCC. That is to say the new character may be more suitable in some speech based psychophysiology information processing field. There are 25 men and 25 women participants, and the range of angle is $\alpha \in (0, \pi)$, with the $0.01\pi$ as the step. There are 2,500 recognition accuracy results in each men set and women set. The distribution of FrCC (with all angle $\alpha$) based man and woman deceptive speech recognition accuracy is presented in the Figs. 7 and 8, respectively. The black solid line is accuracy distribution, and the red dotted line is the average accuracy of MFCC. According to the statistical result, the men groups' average accuracy of MFCC is 56.3%, so approximately 20.7% (518/2,500) of the FrCC based recognition accuracy is higher than 56.3%. The women groups' average accuracy of MFCC is 50.3%, so approximately 60.8% (1,520/2,500) of the FrCC based recognition accuracy is higher than 50.3%.

(E) Each speech frame is regarded as the basic unit for deceptive speech detection under LDA model. It reveals the advantages of FrCC coefficients for this classification task. The vector $w$ is a global optimal vector, but the speech flow is a time-varying process. If the context information can be introduced in the identification system, the speech signal can be mapped onto the state flows. Then, if statistical model are used for classification, there may be a better result.

(F) The speech is a time series signal, and the contextual information may be hidden among the speech frames. The HMM model can mine this information and reveal the relations among the adjacent speech frames. In the HMM recognition system, the men groups' average accuracy of FrCC with best angle is 71.0%, and MFCC is 66.4%. The
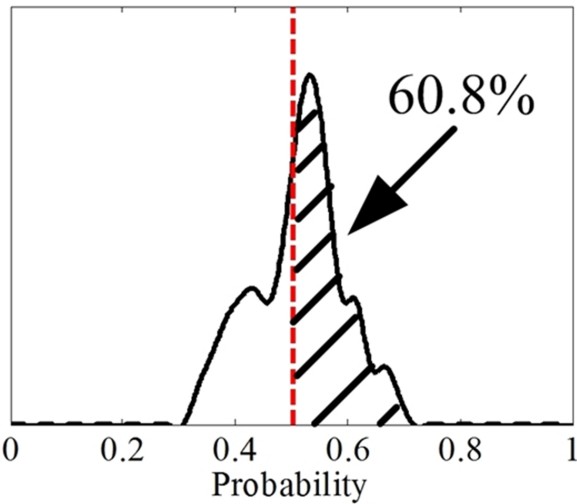

**Figure 8** The distribution of FrCC based deceptive speech recognition accuracy by LDA in women set.

women groups' average accuracy of FrCC with best angle is 70.2%, and MFCC is 65.0%. When the HMM model is involved, the average accuracy is increased by 11.1% and 14.0%, respectively, in two groups. The highest accuracy of FrCC is 82.0% in the men set, and 85.4% in the women set. The largest individual accuracy increase is 31.2% in the women set (from 39.9% to 71.1%, 13th woman) and 18.5% in the men set (from 45.5% to 64.0%, 12th man). The FrCC based deceptive detection accuracy comparison between LDA and HMM are shown in Figs. 9 and 10.

(G) The ROC curve (*Fawcett, 2006*) is usually used to analysis the performance of the identification system. Here, the deceptive detection is a binary classification problem, in which the outcomes are labeled either as positive (*p*) or negative (*n*). There are only four outcomes from a binary classifier, the true positive (TP), false positive (FP), true negative (TN) and false negative (FN). Therefore, we select two parameters, the true positive rate (TPR, sensitivity) and true negative rate (TNR, specificity) to evaluate the performance of the LDA and HMM model. The sensitivity defines how many correct positive results occur among all positive samples during the experiments. Specificity defines how many correct negative results occur among all negative samples during the experiment. The definition equations are shown in (31) and (32). The statistical results are presented from Figs. 11–14. The difference between the sensitivity and specificity of every participant is not large, so LDA and HMM are the suitable tool for dividing the normal or deceptive speech.

$$sensitivity = \frac{TP}{TP + FN} \tag{31}$$

$$specificity = \frac{TN}{FP + TN}. \tag{32}$$

In summary, the experimental results at least reflected that the acoustic feature is effective for lie detection. Faint difference between truthfulness and deceptiveness can be
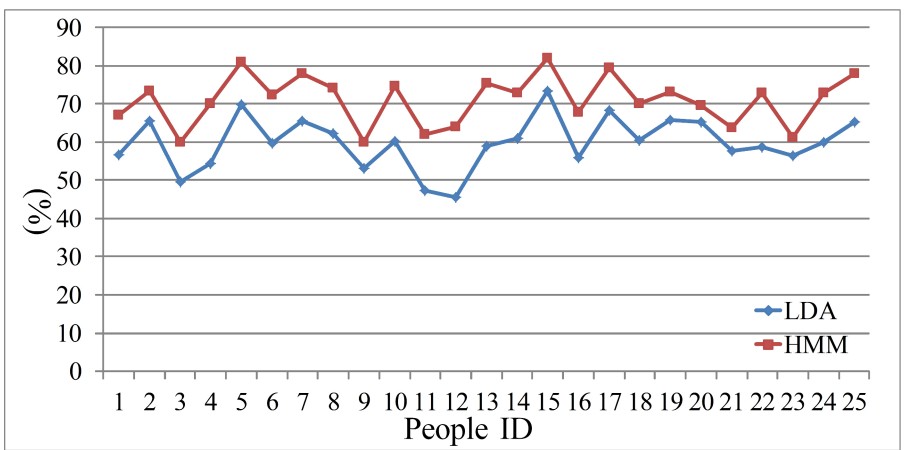

**Figure 9 The FrCC detection accuracy of LDA and HMM in men set.**

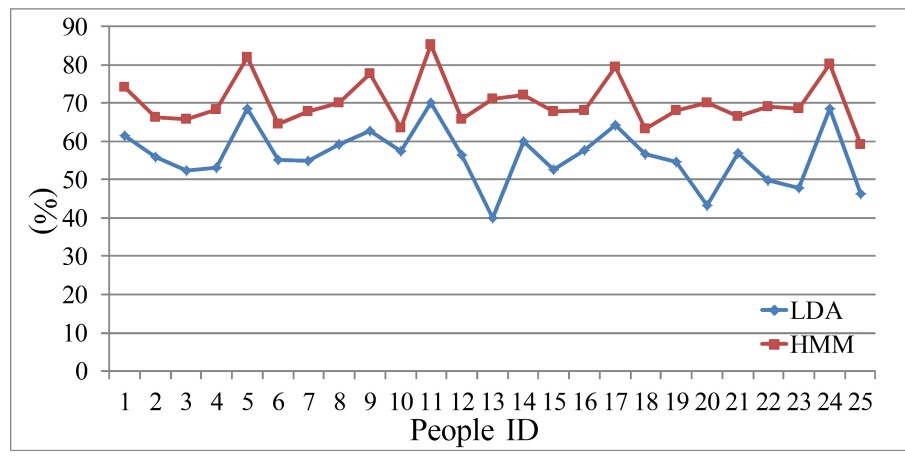

**Figure 10 The FrCC detection accuracy of LDA and HMM in women set.**

expanded under some improved acoustic features such as FrCC, and these characteristics may play an important role in deceptive speech identification.

## CONCLUSION AND PROSPECT

Lie detection based on Speech signal analysis is affected by many factors, such as the psychological quality of the subjects, the way of speaking, interference of environment, and the cost of being exposed, etc. So the development of this technology is relatively difficult. The lack of psychological and physiological research basis also makes less progress in this field. In this paper, fractional Mel cepstral coefficient (FrCC) has been proposed as the speech feature, linear discriminant analysis model (LDA) and hidden Markov model (HMM) are introduced as the classifier. The experiment results show

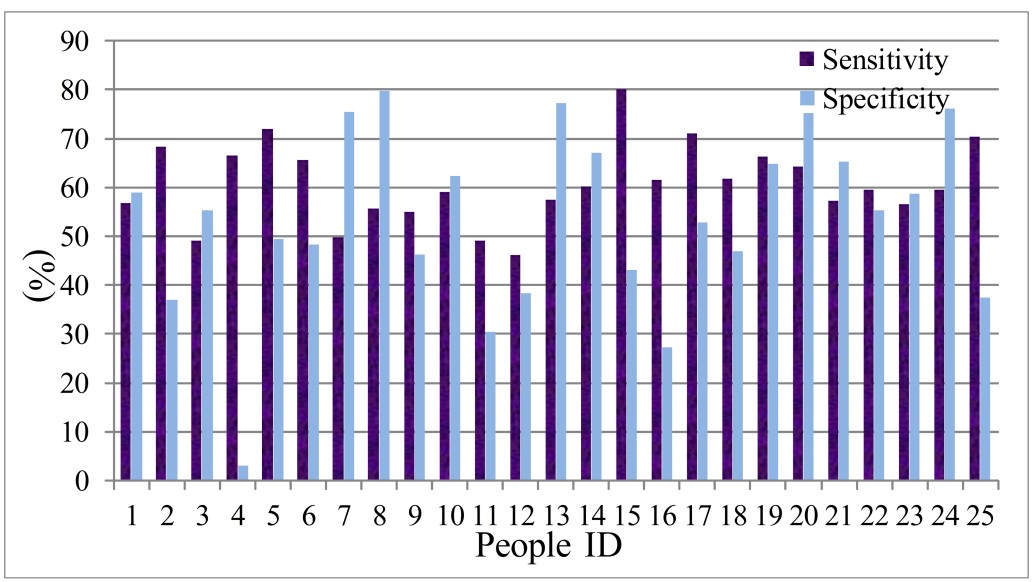

**Figure 11  The sensitivity and specificity of men set in LDA model.**

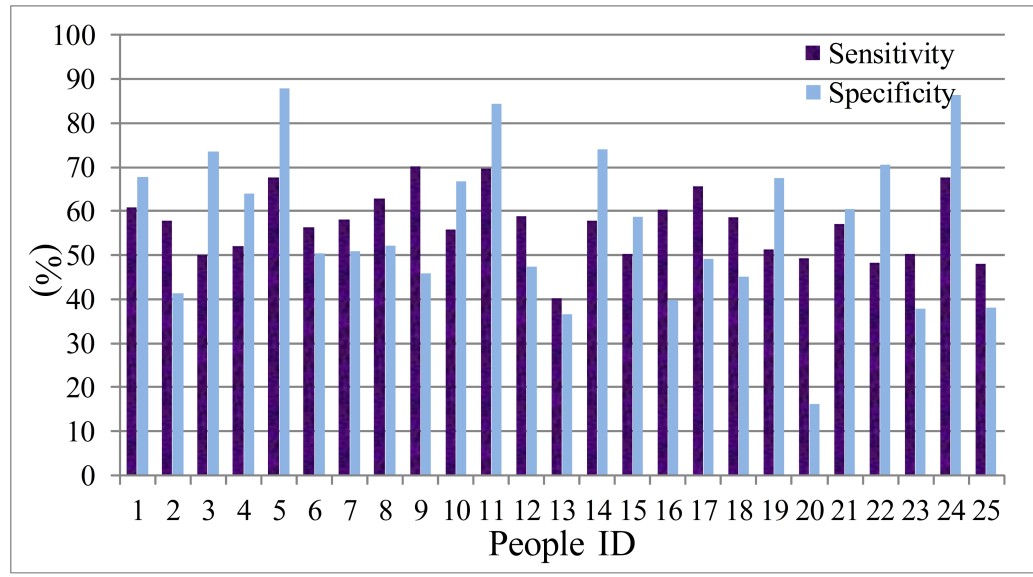

**Figure 12  The sensitivity and specificity of women set in LDA model.**

that the clustering effect of FrCC under optimal angles is better than that of MFCC, and the truthfulness/deceptiveness identification accuracy of FrCC is higher than that of MFCC through LDA or HMM. The successful application has demonstrated that the FrCC parameter can be used in deceptive speech detection, and provides some further experiment evidence in this field.

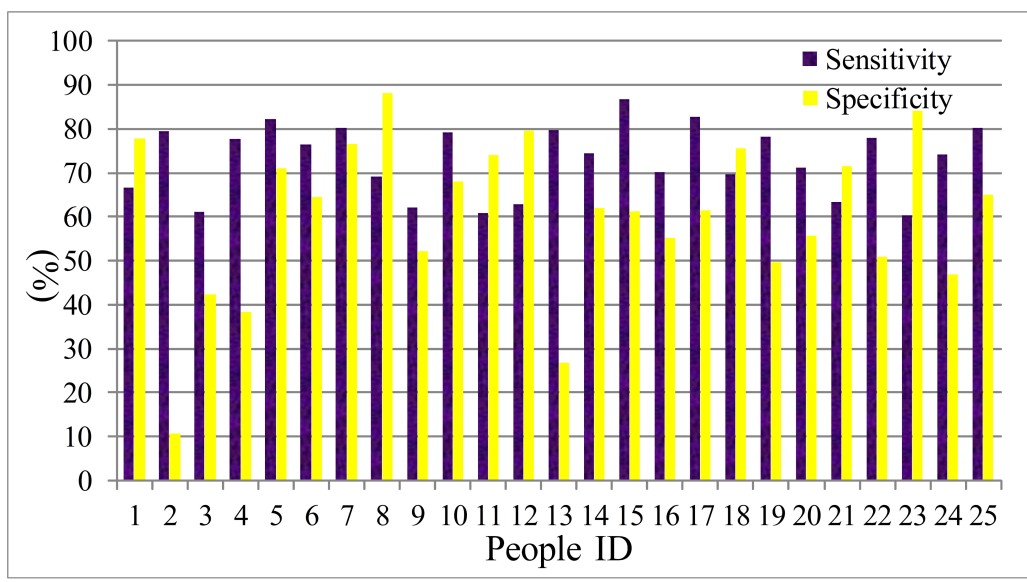

**Figure 13  The sensitivity and specificity of men set in HMM model.**

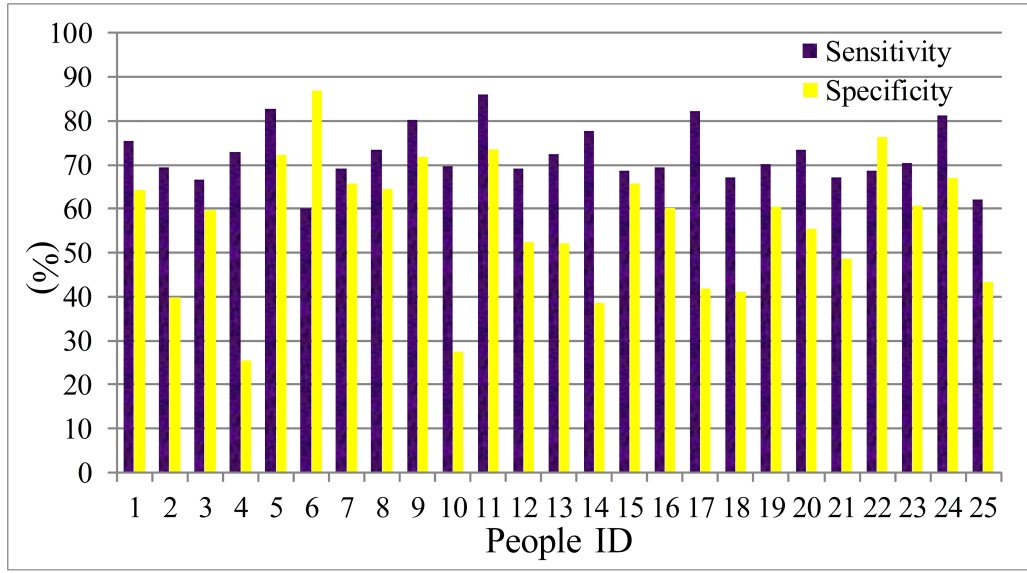

**Figure 14  The sensitivity and specificity of women set in HMM model.**

Future work should mainly focuses on the following aspects: First, to establish a unified optimal angle search mechanism, and achieve complete extraction algorithm of FrCC; Second, to further deep mining-related features, construct a data fusion model, enhance the useful property, and compress the redundant information and interference; Third, the deep mining the time series model, and enhance the contextual information for deceptive speech detection. Speech-based deceptive detection may be an important aid for traditional neuroimaging methods.

### Funding

This work is supported by the National Natural Science Foundation of China (Grant No. 61071215, 61372146), the Postgraduate Research Innovation Project of Jiangsu Province, China (Grant No. CXZZ12_0815), the Natural Science Foundation of Jiangsu Province, China (Grant No. BK20131196, BK20130324, BK2012166), Specialized Research Fund for the Doctoral Program of Higher Education (SRFDP) (Grant No. 20123201120009), and Natural Science Foundation of the Jiangsu Higher Education Institutions of China (Grant No. 12KJB510029). The funders had no role in study design, data collection and analysis, decision to publish, or preparation of the manuscript.

### Grant Disclosures

The following grant information was disclosed by the authors:
Natural Science Foundation of China: 61071215, 61372146.
Postgraduate Research Innovation Project of Jiangsu Province, China: CXZZ12_0815.
Natural Science Foundation of Jiangsu Province, China: BK20131196, BK20130324, BK2012166.
Specialized Research Fund for the Doctoral Program of Higher Education (SRFDP): 20123201120009.
Natural Science Foundation of the Jiangsu Higher Education Institutions of China: 12KJB510029.

### Competing Interests

The authors declare there are no competing interests.

### Author Contributions

- Xinyu Pan conceived and designed the experiments, performed the experiments, analyzed the data, contributed reagents/materials/analysis tools, wrote the paper, prepared figures and/or tables, reviewed drafts of the paper.
- Heming Zhao conceived and designed the experiments, reviewed drafts of the paper.
- Yan Zhou performed the experiments, analyzed the data, contributed reagents/materials/analysis tools, reviewed drafts of the paper.

### Human Ethics

The following information was supplied relating to ethical approvals (i.e., approving body and any reference numbers):

This research was approved by the Institutional Review Boards of Soochow University School of Electronics and Information Engineering, and Suzhou University of Science and Technology School of Electronics and Information Engineering. The speech set recording is carried out in a game style, so all the participants are confirmed with verbal consent.

## Data Availability

The following information was supplied regarding the deposition of related data:

Speech data set can be downloaded here: http://pan.baidu.com/s/1kT69sqV.

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
