# Peer review of "The application of fractional Mel cepstral coefficient in deceptive speech detection"

_PeerJ, doi:10.7717/peerj.1194_

## Round 0.1 · original submission · Major Revisions

Please take attention on the 60% detection rate, and cite 3-5 references showing that this detection rate is acceptable in the field.

Reviewer 1 ·

Basic reporting

This paper examined whether deceptive information can be effectively detected from speech signal. They used new methods to analysis speech signal: Fractional Mel Cepstral Coefficient and LDA Model, which is the novel part of this paper.However, this paper is not well written and well-constructed. The authors should ask a native English speaker to revise the paper. The introduction seemed redundant. And I am not quite clear about the major questions the authors aimed to answer. In addition, there is no discussion part for this paper.
The detection rate in this paper is only less than 60%. In my opinion, this detection rate is not better than chance. Thus, though the authors adopted new methods, but theirs results failed to the validity of this methods. From the research of lie detection perspective, this paper did not make contributions.

Page 1 (line 4 in the introduction). The P300-based lie detector does not reflect the process of psychological and physiological changes during lying. In fact, the P300-based CIT is to detect memory, not deception. Some fMRI-based lie detector also relies on detecting crime-relevant memories.
Page 1 (line 11 in the introduction). The author just said speech signals may provide the basis for lie detection. They should state this more specifically. For example, what advantage of speech signals have in compare to other indices like P300 or fMRI.

Experimental design

The author should state clearly about what protocol they used in this paper. The Concealed information test (CIT) is widely used, because it has a good theoretical foundation. Did the author just compare the deception condition with truthful condition? What is the theoretical foundation on this comparison? The author should also state clearly.

Page 7 (line 4 in speech database) the author said the persons in group-B can ask all kinds of question according to the story. Did every person in group B ask the same questions? How many questions are there in your experiment? How did the author make sure every participant in group A give the same answer for each question? Because you need keep the answers all the same to control the confounding that different answers made.

Validity of the findings

The detection rate in this paper is only less than 60%. In my opinion, this detection rate is not better than chance. Thus, though the authors adopted new methods, but theirs results failed to the validity of this methods. From the research of lie detection perspective, this paper did not make contributions.

When doing the individual analysis, both Ben Shakhar and Rosenfeld typically used ROC curve. This paper should also do the ROC curve to see the detection rate so that your results can be compared with other studies.

Additional comments

No comments.

Reviewer 2 ·

Basic reporting

This is an very interesting research work. Though the accuracy and sensitivity of the proposed techniques are significantly low, it somehow shows some useful information.
A further study that combine neuroimaging method with the proposed techniques may be conducted to improve the lie detection accuracy.

Experimental design

The design is too simple. A more complex and long-term tests should be implemented to show if the present protocol makes more sense.

Validity of the findings

The average accuracy is low, which means the method is more case-dependent. A more detailed and deep discussion on the analysis method, experimental design, accuracy (why it is low) and the future research directions should be added in the revised manuscript.

Additional comments

They claimed that the present method can overcome the job from neuroimaging techniques. We think it can never replace the neuroimaging methods because of the low accuracy.
Further tools should be developed to improve the accuracy of this technique.

---

## Round 0.2 · accepted · Accept

Thanks for clarifying the low detection rate with the literature reference.

Reviewer 2 ·

Basic reporting

Reasonable.

Experimental design

Have flaws but feasible.

Validity of the findings

It is OK.

Additional comments

Need combine neuroimaging techniques with speech detection to improve the sensitivity of the developed method.